# Gender-Specific Patterns of Injury in Older Adults After a Fall from a Four-Wheeled Walker (Rollator): Retrospective Study from a Swiss Level 1 Trauma Centre

**DOI:** 10.3390/ijerph22020143

**Published:** 2025-01-22

**Authors:** Jolanta Klukowska-Rötzler, Fabian Graber, Aristomenis K. Exadaktylos, Mairi Ziaka, Dominik A. Jakob

**Affiliations:** 1Department of Emergency Medicine, Inselspital, Bern University Hospital, Bern University, 3010 Bern, Switzerlandaristomenis.exadaktylos@insel.ch (A.K.E.); mairi.ziaka@gmail.com (M.Z.); dominik.jakob@extrern.insel.ch (D.A.J.); 2Centre of Geriatric Medicine and Rehabilitation, Kantonsspital Baselland, 4101 Bruderholz, Switzerland; 3Department of Visceral Surgery, Lindenhofspital, 3001 Bern, Switzerland

**Keywords:** geriatric patients, four-wheeled walker, rollator, falls, trauma

## Abstract

Aim: As the population is aging, falls by older people, in particular falls from four-wheeled walkers (“rollators”), are a growing problem. These falls must be examined by targeted research and interventions that incorporate gender differences. Therefore, this study examined the injury patterns of elderly patients admitted to a tertiary trauma centre in Switzerland after falls from rollators and focussed on gender differences. Methods: This was a retrospective single-centre study for the period from May 2012 to December 2019 which included elderly patients (≥65 years) who had suffered a fall from a rollator. Injury history, patient data, demographic information, and patient outcomes were compared between males and females, with the data sourced from the Ecare patient database, which contains all information related to patient visits and treatment procedures. Results: A total of 152 eligible patients were included in the analysis, with 56.6% hospitalised at our facility and 14.5% transferred to another hospital. The cohort comprised 50 (32.9%) males and 102 (67.1%) females. Males were more prevalent in the 75–84 age group, while females predominated in the 85 and older group, and this difference was statistically significant (*p* = 0.043). Osteoporosis was significantly more common in females (37.3% vs. 10%, *p* ≤ 0.001). Consequently, treatment with vitamin D and/or calcium was also significantly more prevalent among women (29.4% vs. 8%, *p* = 0.003). Most falls occurred at home (22.4%) or in nursing care facilities and rehabilitation centres (22.4%), without gender-based differences (*p* = 0.570). Men were six times more likely than women to sustain injuries when under the influence of alcohol (*p* = 0.002). Fractures to the lower extremities were the most common injuries, accounting for 34.2% of all injuries, with no statistically significant differences between groups (*p* = 0.063). Head injuries occurred in 34.9% of patients, with a trend towards more injuries in males (44% vs. 30.4%, *p* = 0.098). The cumulative rates of fractures to the pelvis, upper extremities, and lower extremities were significantly higher in females (59.8%) than in males (38%), *p* = 0.011. In-hospital mortality was significantly higher in men than in women (12.9% vs. 2.9%, *p* = 0.026). Operative procedures were significantly more common in women (33% vs. 16.3%; *p* < 0.001). Conclusion: Women were more frequently affected by falls related to rollators than men. Most falls occurred at home, in nursing care facilities, or rehabilitation centres, with no significant gender-based differences. There was a trend toward more head injuries in males, while the cumulative fracture rate of the pelvis, upper extremities, and lower extremities was significantly higher in females. In-hospital mortality was more than four times higher in men. These findings may guide the development of gender-specific interventions to reduce rollator-related injuries in the vulnerable elderly population.

## 1. Introduction

As the world’s population continues to age, the incidence of falls among older adults has increased and poses a significant challenge to healthcare systems, social services, and economies. From an economic perspective, the financial impact of falls is significant, encompassing direct medical expenses such as hospitalisation and additional medical costs, as well as indirect costs associated with lost productivity and long-term care expenses [1]. In Switzerland, falls are the fifth most important cause of disease burden [2,3,4] and are associated with material costs of 1.7 billion Swiss Francs (CHF) per year [5]. This amounts to approximately CHF 180 per resident. Between 2013 and 2017, there were 88,270 falls in patients over the age of 65, 76.2% of whom were women. The problem has been recognised and programmes to reduce the risk of falls have been initiated [5].

In recent years, there has been growing recognition of the intricate interplay between various health factors and the risk of falls in this population. Chronic health conditions, such as osteoporosis, arthritis, and cardiovascular diseases, along with age-related declines in vision, balance, and muscle strength, can significantly contribute to the susceptibility of older adults to falls [6,7,8]. Moreover, polypharmacy [9,10], balance deficits [11], and cognitive impairment [12,13] further compound this risk and underscore the multifactorial nature of fall-related incidents. Progressive visual impairment [14,15] is an important trigger of many falls. This impairment manifests itself in the individual’s failure to notice minor obstacles—such as door thresholds—in good time and can result in stumbling. Moreover, advancing muscle weakness significantly contributes to the incidence of falls [16].

Gender-related differences in risk factors, injury mechanisms, and outcomes of falls in the elderly are well documented [17]. For example, a recent study investigating gender disparities in older adults with falls highlighted that women were more likely to suffer falls from the same level and at home. Moreover, fractures were more common in women, whereas head injuries were more frequent in men. This difference may partly explain the higher in-hospital mortality observed in male patients despite operative procedures and longer hospital stays being more common in female patients [18]. Moreover, notable differences in fall risk profiles are observed between men and women. Specifically, factors such as living alone, dependency in instrumental activities of daily living, underweight, cognitive impairment, and a history of falls are independently associated with injurious falls in women, while hypotension, smoking, heart disease, impaired balance, and previous falls are common risk factors in men [19].

The use of walking aids, particularly rollators (or four-wheeled walkers), by the elderly has become increasingly prevalent as a means to enhance mobility and independence [20,21]. It can be inferred that walking aids are predominantly used by individuals at high risk of falling. Consequently, devices such as rollators are likely to play a significant role in fall prevention. However, the patterns and severity of injury after falls with walking aids are poorly researched. A study from the Netherlands analysed 1869 rollator falls in people over 65 years of age and showed that they were associated with significant injuries [22]. Despite the widespread use of rollators as mobility aids, there is still a lack of comprehensive research on the specific health-related risk factors associated with falls in older adults who use these devices. This gap in knowledge highlights the need for rigorous scientific research to elucidate the underlying mechanisms and to identify effective preventive interventions [23]. Existing data on gender-related differences in falls among elderly rollator users are limited, but it can be assumed that such differences exist, particularly in relation to biomechanics, device handling, and comorbidities. Understanding the gender-specific dimensions of falls in this population is essential for developing tailored interventions and preventive strategies to reduce fall-related morbidity and mortality [24]. By addressing the multifaceted interplay between gender, osteoporosis, and other risk factors, healthcare practitioners can optimise efforts to prevent falls and thus promote healthier aging among older adults using rollators [25,26]. Therefore, the aim of this study is to identify differences between older men and women who fall when using a rollator.

## 2. Materials and Methods

### 2.1. Study Design and Setting

A single-centre retrospective study was conducted with data on falls related to rollator use over a seven-year period—from 1 May 2012 to 31 December 2019—at the University emergency department (ED) of the Insel Hospital Bern (Switzerland), a certified Level I adult trauma centre affiliated to the University of Bern.

This retrospective study was conducted in the emergency department (ED) of Insel Hospital, which is a university hospital and a level 1 tertiary trauma centre located in Bern, the capital of Switzerland. This is a top-tier facility specialising in treating patients with severe and complex injuries that require immediate, comprehensive care. In practice, the hospital is equipped with advanced medical resources and specialised personnel and infrastructure, so that it can provide extensive emergency services 24/7. In 2023, the ED of Insel Hospital treated nearly 52,500 patients, of whom 13,980 were admitted for surgery, and 8674 through the fast-track system, under which many minor orthopaedic and surgical cases are also treated.

### 2.2. Data Handling and Extraction

The data are sourced from the routine database of the documentation system “E.Care” (E.care BVBA, ED 2.1.3.0, Turnhout, Belgium) of the ED of Inselspital Bern. Only data required for normal patient care were collected. E.Care is typically classified as an electronic medical record (EMR) system. EMRs are digital versions of patients’ paper charts and are designed to facilitate the management of patient information, to streamline workflows, and to improve the quality of care. This programme was used exclusively in the emergency department during the period of the present study.

A full-text screening was then performed within this dataset to identify patients with trauma after falls related to four-wheeler injuries. We also accounted for spelling variations and errors. Patients over 65 years of age were included in this study by keyword search for “fall” and then full-text analysis of identified hits related to rollator injuries

In accordance with the established inclusion and exclusion criteria, all retrieved results were initially screened by a single investigator. In cases where there was any uncertainty regarding whether to include or exclude a particular record, the investigator consulted with the project leader (senior researcher) to jointly make a final decision. Since only one individual conducted the initial selection, no measure of inter-rater reliability (e.g., Cohen’s kappa) was calculated. The validation process involved discussion and collective analysis of ambiguous cases to ensure consistency with the inclusion and exclusion criteria and to maintain data quality.

### 2.3. Inclusion and Exclusion Criteria

Patients aged 65 years or older who were admitted to the adult ED of the University Hospital of Bern between 1 May 2012 and 31 December 2019 were eligible if their falls could be clearly attributed to the use of a rollator (four-wheeled walker), whether during walking, standing, or sitting on the device. At the University Hospital Bern, the adult ED is designated for patients aged 16 and older. Children and adolescents up to the age of 16 are referred directly to the ED of the children’s hospital. Patients without general data use consent were excluded, as were patients with non-acute injuries—those suffered more than three days before admission to the hospital.

### 2.4. Study Outcomes

The primary outcome was gender-specific differences in the type of injury in patients after falls with a rollator.

### 2.5. Secondary Outcomes

Secondary outcomes included patient demographics, patient admission information (mode of presentation, type of admission, triage (Swiss medical emergencies and disasters with 1–4 range, Glasgow Coma Score on admission)), details of the circumstances of the injury (location of fall, aetiology of fall), comorbidities, medications (psychiatric medications, antiepileptics, opioids, NSAIDs [27,28], cardiovascular medications, insulin, and osteoporosis medications), different groups of anticoagulants, alcohol intoxication, clinical outcomes (hospital admission, 30-day mortality, length of hospital stay—LOS) and hospital costs in Swiss francs (CHF). With regard to alcohol intoxication, it should be noted that this is not measured routinely but only at the discretion of the treating team.

### 2.6. Statistical Analysis

Statistical analyses were conducted using STATA version 18.1 (StataCorp, College Station, TX, USA). The presentation of continuous variables was based on the results of normality testing (Shapiro–Wilk test), with medians (interquartile ranges, IQRs) or means (standard deviations, SDs) displayed accordingly. For the comparison involving two unmatched groups (e.g., males vs. females), p values for continuous variables were derived from the Wilcoxon rank-sum test or the unpaired *t* test for normally and non-normally distributed variables, depending on normality. For categorical variables, the data were presented as counts (percentages) within each category, and *p*-values were calculated using the chi square test for comparisons between unmatched groups.

### 2.7. Ethics

The most recent revision of the principles of the Declaration of Helsinki and Guidelines of Good Clinical Practice was fulfilled [29]. The cantonal (district) ethics committee approved this study under Bern Number b2021-01909. Since our patients were fully anonymised before analysis, in accordance with Swiss law and ethical approval, informed consent was not mandatory. This was a study with retrospective design and all data were anonymised prior to analysis. Because of the use of coded routine care patient data, no informed consent was needed according to Swiss law.

## 3. Results

### 3.1. Data Selection

Between 1 May 2012 and 31 December 2019, a total of 739 patients aged 65 years or older were identified as having sustained fall injuries. A total of 554 patients were excluded because their admission was not linked to the use of a rollator, along with an additional 32 patients who were excluded due to lack of concordance with general consent. Ultimately, 152 patients with rollator-related injuries were included in our study (Figure 1). The number of patients involved in rollator-related accidents corresponds to a mean of two admissions per month, or approximately 0.005% of all cases treated annually in our emergency department.

Of all patients presenting with rollator-related injuries, 32.9% (*n* = 50) were male and 67.1% (*n* = 102) female. Male patients were younger than female patients (median age: male 80.5 years (IQR 75.25–84.75) vs. female 83.0 years (IQR 79–88), *p* = 0.043). The most common age group among male patients was 75–84 years (*n* = 27, 54.0%), whereas for females, it was ≥85 years (*n* = 50, 49.0%) (*p* = 0.021) (Table 1).

No significant differences were observed for the following parameters: year, month, day of the week, time of consultation, triage group, route of admission, and route of discharge (Table 1).

### 3.2. Prior Comorbidities and Pre-Existing Medication

The prevalence of osteoporosis was notably higher in the female group, with 37.3% (*n* = 38) affected, compared to 10% (*n* = 5) in the male group (*p* < 0.001). No statistical differences were observed between men and women with regard to comorbidities (*p* = 0.779).

Psychiatric (55.3%) and cardiovascular drugs (43.4%) were most commonly prescribed, with no statistically significant differences between the two groups (*p* = 0.051 and 0.551, respectively). Overall, 67.8% (*n* = 103) of all patients were receiving antiplatelet therapy or oral anticoagulation. Sixty patients (39.5%) were on antiplatelet therapy, seventeen (11.2%) were taking vitamin K antagonists, and twenty-nine (19.1%) were on direct oral anticoagulants. No statistically significant differences were observed between the studied groups (Table 2). As regards osteoporosis therapies, medications from the group vitamin D and/or calcium were significantly more frequently used by women (29.4%, *n* = 30) than by men (8%, *n* = 4) (*p* = 0.003), whereas other antiresorptive or osteoanabolic therapies for osteoporosis (bisphosphonates, denosumab, teriparatide, and selective oestrogen receptor modulators (SERMs)) did not differ significantly between groups (Table 2).

### 3.3. Site and Aetiology of the Fall

Overall, most of the falls occurred at home (*n* = 34, 22.4%), followed by injuries suffered at a nursing home/assisted living/psychiatric unit/rehabilitation (*n* = 34, 22.4%) and on the street (*n* = 30, 19.7%), without statistical differences between genders (*p* = 0.570). As regards the aetiology of the fall, tripping and falling were noted in 91 patients (59.9%) and syncope in 17 patients (11.2%). No statistical differences were observed between groups (*p* = 0.367). Alcohol intoxication was diagnosed in six male patients (12.0%) and in one female patient (1.0%) (*p* = 0.002).

### 3.4. Pattern of Injury

In total, 53 patients sustained a head injury (34.9%), with a trend toward a higher incidence among male patients (44.0% vs. 30.4%, *p* = 0.098) (Figure 2). More specifically, concussion/contusion of the head was noted in 18 (11.8%) patients, subdural haematoma in 10 (6.6%), subarachnoid haemorrhage in 3 (2.0%), intracerebral bleeding in 8 (5.3%), and skull fracture in 6 patients (3.6%). No statistically significant differences were observed between the groups. Minor skull injuries (injuries of the integument of the skull, such as lacerations) were present in 23 patients (15.1%) in total and were significantly higher among male patients (24.0% vs. 10.8%, *p* = 0.033) (Table 3).

Spinal injuries were identified in a minority of patients (*n* = 12, 7.9%) and included cervical, thoracic, and lumbar spine fractures. No significant differences were detected between the groups studied (*p* = 0.059). Additionally, maxillofacial trauma was noted in a substantial proportion of patients (*n* = 28, 18.4%), with facial bone fractures occurring more frequently in females (19%, *n* = 20) than in males (16%, *n* = 8), albeit without statistically significant differences between the genders (*p* = 0.59). Chest injuries were observed in a small percentage of patients (*n* = 15, 9.9%) and primarily consisted of rib fractures and minor thoracic trauma, with no notable gender differences (Table 3).

Trauma to the pelvis and the upper and lower extremities was observed in a higher percentage of females than in males (Figure 2), with no significant differences between the two groups in terms of the specific types of injuries. However, if all injuries were combined (pelvis, upper extremities, and lower extremities), a significant statistical difference was observed, with females accounting for 59.8% (*n* = 61) of cases, compared to 38% (*n* = 19) of males (*p* = 0.011).

### 3.5. Outcomes

#### 3.5.1. Hospitalisation and Costs

Fifty-nine female (57.8%) and twenty-seven male (54.0%) patients were admitted to our hospital. Fifteen females (14.7%) were transferred to another hospital. The mean hospital stay for women was 4 days (IQR: 0–10), and for men, 3 days (IQR: 0–8), (*p* = 0.258). The median costs per patient were CHF 7084 (IQR CHF 2056–19,041), which is approximately USD 8005.72 (IQR: USD 2323.28–21,516.33). No statistically significant differences (*p* = 0.125) in median costs per patient were observed between males and females (Table 4).

#### 3.5.2. Surgical vs. Non-Surgical Treatment

Operative procedures were significantly more common in females than in males (33 cases, 33% vs. 8 cases, 16.3%; *p* < 0.001), while 85.7% of men and 67% of women underwent conservative treatment.

#### 3.5.3. Mortality

In-hospital mortality was more than four times higher in the male group (12.0% vs. 2.9%, *p* = 0.026). However, 30-day mortality was not statistically different between the two groups (12.0% vs. 7.8, *p* = 0.405) (Table 4).

The causes of death in the studied patients reflect a complex interplay between acute injuries and chronic pre-existing conditions, which significantly influenced their health outcomes. Many patients died as a result of traumatic injuries, such as falls leading to skull fractures, subdural hematomas, spinal cord injuries, and other trauma-related complications. In some cases, these injuries were complicated by post-operative delirium, respiratory insufficiency, and sepsis. In addition to acute injuries, the majority of patients had multiple chronic conditions, including cardiovascular diseases (such as coronary artery disease, hypertension, and heart failure), neurological disorders (including dementia, polyneuropathy, and balance disturbances), and metabolic conditions like diabetes and osteoporosis. These underlying health issues often exacerbated the severity of trauma and infections, making recovery difficult and increasing the risk of fatal outcomes. Several patients experienced complications from pre-existing conditions, such as severe heart failure, kidney disease, and chronic respiratory conditions, which directly contributed to mortality. Infections, particularly pneumonia, urinary tract infections, and Clostridium-induced colitis, were also prevalent and played a significant role in the patients’ deterioration. Additionally, many patients suffered from impaired mobility, either due to their chronic conditions or as a result of falls, further increasing their vulnerability to life-threatening injuries and complications. Ultimately, the data indicate that the mortality of these patients was heavily influenced by both acute events (such as traumatic falls and surgical complications) and the cumulative burden of chronic health conditions. This highlights the importance of comprehensive medical management, particularly for patients with multiple comorbidities, to improve their resilience and reduce the risk of fatal outcomes.

## 4. Discussion

As the population ages, more and more people are at risk of falls [30]. As a consequence, the use of walking aids such as rollators is also increasing. Rollators are designed to enhance mobility, alleviate discomfort, to improve confidence and balance, and finally to reduce the risk of falls [31,32]. Nevertheless, it is important to note that, in addition to the physiological changes associated with aging, older patients may also experience impaired mental status, chronic illnesses, polypharmacy, and environmental hazards, all of which can contribute to severe and potentially related injuries [33]. As a consequence of these many factors that may contribute to falls, gender disparities in fall incidence have also been well documented. Epidemiological studies have also consistently demonstrated differences in fall rates between men and women [34,35,36]. The consequences of falls among elderly rollator users are also gender-specific, and men often experience more severe outcomes than women [37]. As recorded in our study, older adults are exposed to a risk of severe, and potentially fatal, injuries while utilising a four-wheeled walker. Overall, three women and six men died during hospitalisation, corresponding to a relatively high mortality rate of 12% for males and 2.9% for females, which is comparable with the findings of previous studies [22].

A total of 152 rollator-related injuries among senior patients (>65 years) were recorded from 2012 to 2019, averaging approximately two per month. Nearly 70% of patients presenting with rollator-related falls were female, which aligns with previous studies indicating that women, particularly those aged 85 and older, are at significantly higher risk of falling than men [22,38,39]. Previous studies have identified gender-related factors linked to falls, encompassing differences in reaction time, muscle strength, and range of motion of limb joints. Functional limitations were shown to be more prevalent in women, which has been attributed to several factors, including their longer lifespan, increased occurrence of non-fatal chronic conditions, intrinsic factors such as reduced muscle strength and lower bone density, and elevated rates of sedentary behaviour and obesity in their lifestyle [40,41]. In our study, there were no significant differences in the severity of injuries between men and women. This implies that women may have presented to the emergency department with a lower threshold for seeking medical attention. These findings are consistent with our observations.

The mean age on admission was greater for women than for men. Overall, 49% of all female patients and 26% of male patients were over 85 years of age. Women older than 75 years also exhibited a higher prevalence of geriatric risk diagnoses, including gait and balance disorders and cognitive impairment. Additionally, our study revealed a higher prevalence of ophthalmological and psychiatric risk diagnoses among female patients, which suggests that poor vision and psychiatric disorders—with associated medication—may trigger some falls. This finding aligns with prior research that has identified vision impairment as a significant risk factor for falls [42]. Moreover, the association between psychiatric diseases, such as depression, and falls in the elderly is well established. Nevertheless, unlike the study by Gale et al., which identified depression and poor eyesight as significant factors for falls in male geriatric patients but not in females, our study found a higher incidence of psychiatric and ophthalmological diagnoses among female rollator users [36]. In our aging societies, multimorbidity is widespread, and often compels older patients to undergo multiple medication [43]. Moreover, as individuals age biologically, it becomes crucial to acknowledge that alterations in pharmacokinetics and pharmacodynamics are common [44]. Our study revealed that—although this was not statistically significant upon admission—females tended to have a higher number of pre-existing medications (median of 10, with a range of 6 to 13) compared to males (median of 7, with a range of 4 to 11). In our study, psychiatric medications were more commonly prescribed to female patients, a finding that was nearing statistical significance. Ultimately, the utilisation of psychotropic and sedative medications, as frequently prescribed to older adults, correlates with an elevated incidence of falls. These factors constitute modifiable variables in strategies to prevent falls [45,46]. In line with this, our findings are consistent with previous research identifying psychiatric medications as significant risk factors for falls among elderly women [47,48].

In our study, most falls occurred at home or in care facilities (both 22.4%), followed by falls on the street, with no significant gender differences observed. Tripping and falling were the most common aetiology of falls and were noted in over half of the patients, while syncope accounted for a smaller proportion. Alcohol intoxication was diagnosed more frequently in male patients than in female patients. Even though rollators are designed to reduce the risk of falls, safer gait is not consistently associated with greater stability. This inconsistency stems from the difficulty in distinguishing between users who rely on rollators for balance support and those who use them primarily to bear weight [21,49]. Indeed, previous research suggests that backward falls during weight transfer on a rollator are a common cause of falls and are often associated with challenges related to manoeuvrability, lateral stability, and control over wheel velocity. Improvements in the design of rollators should focus on these aspects and may be essential for minimising falls among older adults [50]. Certainly, the analysis of the use of rollators in both laboratory and everyday settings underscores the importance of studying gait in real-world conditions and may lead to novel approaches, such as employing wearable inertial sensors to understand the interaction between the rollator, the user, and the walking environment [51,52].

Head injuries, including concussion/contusion of the head, subdural haematoma, subarachnoid haemorrhage, intracerebral bleeding, and skull fracture, were sustained by 53 patients, with a higher incidence among male patients. Although this failed to reach statistical significance, it may indicate that men are at higher risk for head injuries. Our findings align with Alter and co-workers’ study (2023) on the role of sex in skull fracture risk among patients over 65 conducted at two level one trauma centres serving 360,000 geriatric residents. Their study revealed that males had a significantly higher rate of skull fracture than females (4.6% vs. 3.0%, OR 1.5, 95% CI: 1.2–2.1, *p* = 0.002)—a trend consistent across race/ethnicity and injury mechanisms [53]. The observation in one publication that men over 65 are more likely to sustain head injuries in rollator-related trauma can be interpreted in several ways. One hypothesis is that a higher prevalence of alcohol consumption among older men contributes to these injuries [54]. In our study, six times more men than women were injured under the influence of alcohol (*p* = 0.002). This finding can also be interpreted in light of other circumstances. For example, gender differences in risk-taking behaviour and physical activity levels may play a role. Older men may engage in more physically demanding activities or take more risks when using a rollator, thus increasing their susceptibility to falls and head injuries [55,56]. On the other hand, physiological and anatomical differences between men and women can affect injury patterns. Men may have different gait and balance characteristics, or differences in muscle mass and distribution, which may affect the dynamics of falls [57]. Men suffered more head injuries in our study, possibly due to a combination of medical and environmental factors. Firstly, they received oral anticoagulation more frequently (72% vs. 65.7%), and this increases the risk of severe bleeding [58,59]. Moreover, men experienced significantly higher rates of alcohol intoxication (12.0% vs. 1%, *p* = 0.002), which might result in a loss of protective reactions and facilitate the higher frequency of head injuries. Furthermore, men were more likely to fall in public places (24% vs. 17.6%), while women predominantly fell at home or in nursing and rehabilitation facilities, which are typically considered to be safer environments [56]. The fact that men experience a more severe pattern of head injury may explain the significantly higher in-hospital mortality observed among men in our study.

While men had a higher incidence of head injuries, their more severe injury patterns probably contributed to the increased mortality rate. Traumatic brain injuries, particularly in individuals on anticoagulants or under the influence of alcohol, carry a higher risk of complications such as intracranial haemorrhage, brain oedema, or respiratory failure, which can be fatal if they are not promptly managed. In our study, the higher mortality in men can be attributed to a combination of these risk factors. Anticoagulation therapy increases the likelihood of major bleeding after head trauma, while alcohol intoxication worsens injury severity and impairs timely medical intervention. These factors, along with falls in public spaces, probably lead to more severe brain injuries and may overwhelm medical resources and raise mortality. This finding highlights the need for tailored management strategies, including more vigilant screening for patients on anticoagulants or with alcohol intoxication and interventions to reduce falls in public spaces. If clinicians bear in mind that the risk of fatal outcomes is greater in men, this could encourage them to prioritise male patients. Recognising the higher risk of fatal outcomes in men can guide clinicians in prioritising early intervention and intensive care and ultimately improve survival rates for this high-risk group [18].

Injuries to the upper extremities were more commonly observed in females than in males. Radius fractures, in particular, occurred more frequently in women, and fractures of the humerus and ulna were also more common among females, although these differences were not statistically significant. Injuries to the lower extremities were more evident in females than in males, although no statistically significant differences were observed. Femur fractures were the most common injury to the lower extremities in women, followed by fractures to the fibula and tibia. As a consequence, the cumulative fracture rate of pelvis, upper extremities and lower extremities was significantly higher in females than in males. These findings are in line with the literature. Fragility fractures commonly occurred at sites such as the hip, distal forearm, vertebrae, and humerus, and our findings are consistent with previous research showing that, in developed nations, these fractures may affect approximately one in three women and one in five men aged 50 years or older throughout their remaining lifespan [60]. Moreover, it is well established that low-energy trauma resulting from falls at standing height is the primary cause of radius injury in older adults, with adverse health outcomes associated with advanced age, the female gender, and compromised bone healing [61]. Indeed, reduced bone mineral density not only increases the likelihood of fractures but also involves significant physical, psychological, social, and financial implications for the affected individuals [62,63]. Indeed, in our study, osteoporosis was documented in a notable portion of patients, with a significantly higher prevalence in women than in men. In addition, a significantly greater proportion of female patients with osteoporosis than male ones were treated with vitamin D and/or calcium. As geriatric patients, especially women, face a high risk of osteoporosis and fractures, all older adults should be educated on habits that support bone health, including appropriate exercise and quitting smoking as appropriate. It is crucial to perform annual fall assessments and develop strategies like fall prevention education, which can be addressed in primary care settings, assisted living facilities, or even emergency departments. Moreover, health care providers must fully understand the potential risks and benefits of diagnosing and treating osteoporosis in the older senior population, as this can contribute to falls and fractures, particularly in women [64]. The prevalence of osteoporosis in the EU is estimated at 5.6%. In Switzerland, about 22.6% of women and 6.6% of men aged 50 and over had osteoporosis in the same period, a total of 524,000 people [65]. As well as pain and disability, some fractures are also associated with premature mortality. An international study showed that the number of fracture-related deaths varies between EU countries, and this reflects the incidence of fractures rather than the standard of care. Hip fractures are the most serious consequence of osteoporosis in terms of morbidity, mortality, and health care costs. The remaining lifetime probability of hip fracture (%) at age 50 for men and women was 7.1% and 22.5%, respectively, placing Switzerland in the top tertile of risk for men and women [65].

In our study, significantly more females underwent an operative procedure. These findings are consistent with previous research showing that elderly female patients are more likely to undergo surgery during hospitalisation following a fall [18]. This could be due to the higher incidence of fractures requiring surgical intervention among women, as well as the increased prevalence of osteoporosis in females, which leads to more severe injuries compared to men.

## 5. Limitations

While the present study possesses several strengths, it is important to acknowledge certain limitations. Documentation bias may persist in any retrospective study, despite a thorough review of all included data. Moreover, as this study is retrospective in nature, some data may be missing, despite efforts to ensure comprehensive data extraction and to minimise missing values. However, it is anticipated that these biases are uniformly distributed across both patient groups and are therefore unlikely to impact this study’s findings. Furthermore, our research incorporates data from patients who visited the ED of a tertiary hospital. Consequently, it is likely that our results primarily reflect the effects of severe injuries among rollator users. Patients who received full treatment in primary care were thus excluded. Finally, our study lacked long-term follow-up data, which could have offered further insights into the duration of disability and health costs.

## 6. Conclusions

Women were more frequently affected by rollator-related falls than men. Most falls occurred at home, in nursing facilities, or in rehabilitation centres, with no significant gender-based differences. There was a trend towards more head injuries in males, whereas the cumulative fracture rate of pelvis, upper extremities, and lower extremities was significantly higher in females. In-hospital mortality was more than four times higher in male patients. These findings underscore the importance of considering gender-specific differences in injury profiles when evaluating trauma patients after rollator-associated falls. Preventive measures and comprehensive healthcare strategies that are tailored to the unique needs of both men and women should be implemented, as they are crucial to enhancing safety and reducing the incidence of injuries in this vulnerable population.

## Figures and Tables

**Figure 1 ijerph-22-00143-f001:**
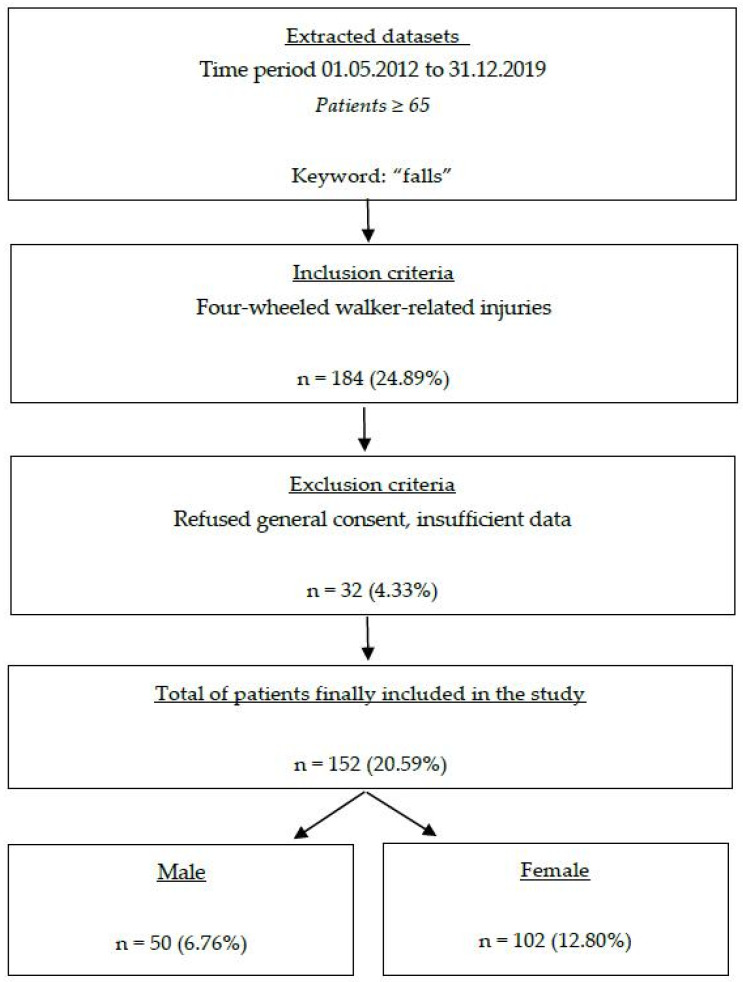
Flow chart illustrating the exclusion and inclusion procedure in this study. The percentage was always calculated in relation to the number of searched records (*n* = 739).

**Figure 2 ijerph-22-00143-f002:**
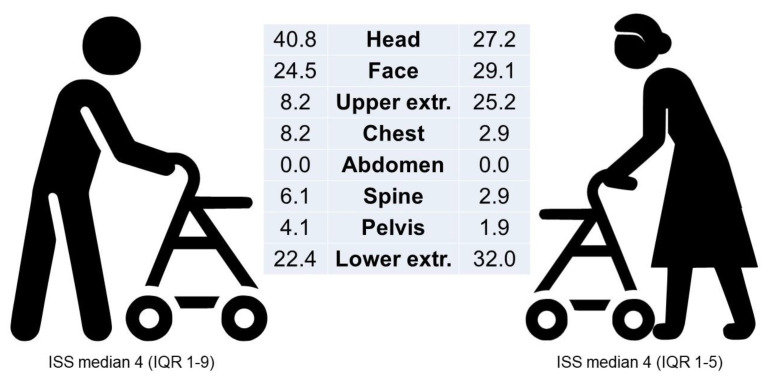
Gender differences in the anatomical location of injuries after rollator-related injuries.

**Table 1 ijerph-22-00143-t001:** Consultation and patient characteristics with gender comparison. The table presents the absolute numbers of patients, with their percentage shares shown in parentheses relative to the total number of cases studied. * = statistically significant values.

Variables	Total (*n* = 152)	Male (*n* = 50)	Female (*n* = 102)	*p*
DEMOGRAPHICS				
Age group				0.021 *
65–74	27 (17.8)	10 (20.0)	17 (16.7)	
≥75–84	62 (40.8)	27 (54.0)	35 (34.0)	
≥85	63 (41.4)	13 (26.0)	50 (40.0)	
Time of presentation				0.792
00.00–02.59	9 (5.9)	2 (4.0)	7 (6.9)	
03.00–05.59	6 (3.9)	2 (4.0)	4 (3.9)	
06.00–08.59	6 (3.9)	2 (4.0)	4 (3.9)	
09.00–11.59	22 (14.5)	9 (18.0)	13 (12.7)	
12.00–14.59	34 (22.4)	14 (28.0)	20 (19.6)	
15.00–17.59	33 (21.7)	11 (22.0)	22 (21.6)	
18.00–20.59	27 (17.8)	7 (14.0)	20 (19.6)	
21.00–23.59	15 (9.9)	3 (6.0)	12 (11.8)	
Route of presentation				0.107
Ambulance	96 (63.2)	37 (74.0)	59 (57.8)	
Self-admission to the hospital	16 (10.5)	2 (4.0)	14 (13.7)	
Family doctor or urgent care centre	13 (8.6)	5 (10.0)	8 (7.8)	
External hospital or psychiatric centre	18 (11.8)	4 (8.0)	14 (13.7)	
REGA (Swiss Air Ambulances)	1 (0.7)	1 (2.0)	0 (0.0)	
Other/no information	8 (5.3)	1 (2.0)	7 (6.9)	
Type of admission				0.068
Walk-in	16 (10.5)	2 (4.0)	14 (13.7)	
(Helicopter) Emergency Services, medical referral	128 (84.2)	47 (94.0)	81 (79.4)	
No information/other	8 (5.3)	1 (2.0)	7 (6.9)	
Triage				0.854
Acute life-threatening	9 (5.9)	2 (4.0)	7 (6.9)	
High urgency	63 (41.4)	22 (44.0)	41 (40.2)	
Urgency	77 (50.7)	25 (50.0)	52 (51.0)	
Less urgency	2 (1.3)	1 (2.0)	1 (1.0)	
Missing information	1 (0.7)	0 (0.0)	1 (1.0)	
Glasgow Coma Score (on entry)	15 (15; 15)	14 (15; 15)	15 (15; 15)	0.991
Site of the fall				0.570
Home or apartment of elderly patient	34 (22.4)	10 (20.0)	24 (23.5)	
Nursing home, assisted living, psychiatric unit, rehabilitation	34 (22.4)	8 (16.0)	26 (25.5)	
On the street/in public	30 (19.7)	12 (24.0)	18 (17.6)	
On public transport	4 (2.6)	1 (2.0)	3 (2.9)	
Other/no information	50 (32.9)	19 (38.0)	31 (30.4)	
Aetiology of the fall				0.367
Tripping and falling	91 (59.9)	27 (54.0)	64 (62.7)	
Syncope	17 (11.2)	8 (16.0)	9 (8.8)	
Other/no information	44 (28.9)	15 (30.0)	29 (28.4)	
Alcohol intoxication	7 (4.6)	6 (12.0)	1 (1.0)	0.002 *

**Table 2 ijerph-22-00143-t002:** Medians and IQR for number of comorbidities and number of drugs, otherwise means and SDs. * = significant differences. The table presents the absolute numbers of patients, with their percentage shares shown in parentheses relative to the total number of cases studied. “Risk diagnosis” refers to specific comorbidities or medical conditions that could potentially increase the risk of complications and adversely affect a patient’s rehabilitation or clinical outcome.

Variables	Total (*n* = 152)	Male (*n* = 49)	Female (103)	*p*
COMORBIDITIES				
Number of comorbidities	6 (3; 9)	6 (4; 10)	6.5 (3; 9)	0.779
Musculoskeletal risk diagnosis	57 (37.5)	19 (38)	38 (37.3)	0.929
Internal risk diagnosis	117 (77.0)	42 (84.0)	75 (73.5)	0.150
Neurological risk diagnosis	43 (28.3)	15 (30.0)	18 (27.5)	0.743
Psychiatric risk diagnosis	23 (15.1)	6 (12.0)	17 (16.7)	0.451
Ophthalmological risk diagnosis	28 (18.4)	6 (12.0)	22 (21.6)	0.153
Otolaryngology risk diagnosis	14 (9.2)	5 (10.0)	9 (8.8)	0.814
Geriatrics risk diagnosis	64 (42.1)	21 (42.0)	43 (42.2)	0.985
MEDICATION				
No of drugs	9 (3; 13)	7 (4; 11)	10 (6; 13)	0.082
On psychiatric medication	84 (55.3)	22 (44.0)	62 (60.8)	0.051
On antiepileptics	15 (9.9)	3 (6.0)	12 (11.8)	0.263
On opioids	42 (27.6)	11 (22.0)	31 (30.4)	0.277
On NSAID	7 (4.6)	0 (0.0)	7 (6.9)	0.058
On cardiovascular drugs	66 (43.4)	20 (40.0)	46 (45.1)	0.551
On insulin	11 (7.2)	4 (8.0)	7 (6.9)	0.799
Anticoagulation	103 (67.8)	36 (72.0)	67 (65.7)	0.434
PAI	60 (39.5)	22 (44.0)	38 (37.3)	0.424
Vitamin K antagonists	17 (11.2)	4 (8.0)	13 (12.7)	0.383
DOAC	29 (19.1)	11 (22.0)	18 (17.6)	0.521
Heparin/LMWH	1 (0.7)	0 (0.0)	1 (1.0)	0.482
Osteoporosis	43 (28.3)	5 (10.0)	38 (37.3)	<0.001 *
Osteoporosis treated with antiresorptive or osteoanabolic drugs	15 (9.9)	3 (6.0)	12 (11.8)	0.263
Osteoporosis treated with vitamin D and/or calcium	34 (22.4)	4 (8.0)	30 (29.4)	0.003 *
Osteopenia	4 (2.6)	1 (2.0)	3 (2.9)	0.733
Osteopenia treated with vitamin D and/or calcium	2 (1.3)	0 (0.0)	2 (2.0)	0.319

Abbreviations: NSAID: non-steroidal anti-inflammatory drug; PAI: platelet aggregation inhibitor; DOAC: direct oral anticoagulant; LMWH: low-molecular-weight heparin.

**Table 3 ijerph-22-00143-t003:** Gender-related anatomical location and severity in patients admitted after injuries related to four-wheel rollator. * = statistically significant. The table presents the absolute numbers of patients, with their percentage shares relative to the total number of cases studied shown in parentheses.

Variables	Total (*n* = 152)	Male (*n* = 49)	Female (*n* = 103)	*p*
INJURIES				
INTRACRANIAL BLEEDING				
Subdural haematoma	10 (6.6)	6 (12.0)	4 (3.9)	0.059
Epidural haematoma	2 (1.3)	0 (0.0)	2 (2.0)	0.319
Subarachnoid haemorrhage	3 (2.0)	1 (2.0)	2 (2.0)	0.987
Intracerebral bleeding	8 (5.3)	3 (6.0)	5 (4.9)	0.776
Minor skull injury	23 (15.1)	12 (24.0)	11 (10.8)	0.033 *
HEAD INJURY				
Head injury, overall	53 (34.9)	22 (44.0)	31 (30.4)	0.098
Concussion/contusion of head	18 (11.8)	5 (10.0)	13 (12.7)	0.623
Skull fracture (vault, base)	6 (3.6)	2 (4.0)	4 (3.9)	0.981
Intracranial bleeding	0 (0.0)	0 (0.0)	0 (0.0)	-
FACE				
Face injury, overall	28 (18.4)	8 (16.0)	20 (19.6)	0.590
Facial bone fractures	15 (9.9)	5 (10.0)	10 (9.8)	0.970
Minor facial injury	17 (11.2)	5 (10.0)	12 (11.8)	0.746
CHEST INJURY				
Chest injury, overall	15 (9.9)	5 (10.0)	10 (9.8)	0.970
Haemothorax	0 (0.0)	0 (0.0)	0 (0.0)	-
Pneumothorax	1 (0.7)	1 (2.0)	0 (0.0)	1.152
Rib fracture	2 (1.3)	0 (0.0)	2 (2.0)	0.319
Multiple rib fractures (≥3 ribs)	3 (2.0)	1 (2.0)	2 (2.0)	0.987
Sternum fracture	2 (1.3)	1 (2.0)	1 (1.0)	0.604
Minor chest injury	8 (5.3)	3 (6.0)	5 (4.9)	0.776
SPINE				
Spine injury, overall	12 (7.9)	1 (2.0)	11 (10.8)	0.059
Cervical spine fracture	4 (2.6)	0 (0.0)	4 (3.9)	0.156
Thoracic spine fracture	4 (2.6)	0 (0.0)	4 (3.9)	0.156
Lumbar spine fracture	2 (1.3)	0 (0.0)	2 (2.0)	0.319
Minor spine injury (contusion)	3 (2.0)	1 (2.0)	2 (2.0)	0.987
PELVIS				
Pelvis injury, overall	5 (3.3)	0 (0.0)	5 (4.9)	0.111
Stable pelvic ring fracture (Type A)	2 (1.3)	0 (0.0)	2 (2.0)	0.319
Unstable pelvic ring fracture (Types B and C)	3 (3.0)	0 (0.0)	3 (2.9)	0.221
Acetabulum fracture	1 (0.7)	0 (0.0)	1 (1.0)	0.482
UPPER EXTREMITIES				
Injury to the upper extremities, overall	36 (23.7)	9 (18.0)	27 (26.5)	0.248
Clavicle fracture	2 (1.3)	0 (0.0)	2 (2.0)	0.319
Scapula fracture	1 (0.7)	0 (0.0)	1 (1.0)	0.482
Humerus fracture	7 (4.6)	1 (2.0)	6 (5.9)	0.283
Radius fracture	5 (3.3)	1 (2.0)	4 (3.9)	0.533
Ulna fracture	2 (1.3)	0 (0.0)	2 (2.0)	0.319
Hand fracture	2 (1.3)	0 (0.0)	2 (2.0)	0.319
Minor injury to the upper extremities	22 (14.5)	7 (14.0)	15 (14.7)	0.907
LOWER EXTREMITIES				
Injury to the lower extremities, overall	52 (34.2)	12 (24.0)	40 (39.2)	0.063
Femur fracture	21 (13.8)	6 (12.0)	15 (14.7)	0.650
Tibia fracture	3 (2.0)	0 (0.0)	3 (2.9)	0.221
Fibula fracture	2 (1.3)	0 (0.0)	2 (2.0)	0.319
Patella fracture	1 (0.7)	0 (0.0)	1 (1.0)	0.482
Foot fracture	1 (0.7)	0 (0.0)	1 (1.0)	0.482
Minor injuries to the lower extremities	25 (16.4)	6 (12.0)	19 (18.6)	0.300
LIMB AND HIP FRACTURES ^#^				
Pelvis, upper extremities, and lower extremities	80 (52.5)	19 (38.0)	61 (59.8)	0.011 *

^#^ Stable pelvic fracture, unstable pelvic fracture, acetabulum fracture, clavicle fracture, scapula fracture, humerus fracture, radius fracture, ulna fracture, hand fracture, femur fracture, tibia fracture, fibula fracture, patella fracture, foot fracture. Patients were classified as having zero fractures if they had minor injuries (skin tears and bruises all over the body, joint dislocations) and were then excluded from the cumulative analysis.

**Table 4 ijerph-22-00143-t004:** Outcomes and case costs in patients admitted after rollator-related injuries. Medians and IQR; otherwise means and SDs. The table presents the absolute numbers of patients, with their percentage shares shown in parentheses relative to the total number of cases studied. * = statistically significant values.

Outcome	Total (*n* = 152)	Male (*n* = 49)	Female (*n* = 103)	*p*
Length of hospitalisation (days)	4 (0; 9)	3 (0; 8)	4 (0; 10)	0.258
Cost of hospitalisation (CHF)	7084 [2056; 19,041]	5529 [1772; 13,086]	9139 [2208; 22,119]	0.125
on conversion to U.S. dollars	8005.72 [2323.28; 21,516.33]	6247.77 [2003.36; 14,787.18]	10,327.07 [2495.04; 24,994.47]	
Discharge				
Home	44 (28.9)	16 (32.0)	28 (27.5)	
Admission to the hospital	86 (56.6)	27 (54.0)	59 (57.8)	
Transferred to other hospital	9 (5.9)	7 (14.0)	15 (14.7)	0.844
In-hospital mortality	9 (5.9)	6 (12.0)	3 (2.9)	0.026
30-day mortality	14 (9.2)	6 (12.0)	8 (7.8)	0.405
Treatment				
Conservative	111 (73.1)	42 (85.7)	69 (67.0)	<0.001 *
Operative	41 (26.9)	8 (16.3)	33 (33.0)	

## Data Availability

The data presented in this study are available on request from the corresponding author.

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
