# Peer review of "Gender-Specific Patterns of Injury in Older Adults After a Fall from a Four-Wheeled Walker (Rollator): Retrospective Study from a Swiss Level 1 Trauma Centre"

_ijerph, 2025, doi:10.3390/ijerph22020143_

Round 1

Reviewer 1 Report

Comments and Suggestions for Authors

Dear Editor

Thank you very much for the opportunity to review this interesting manuscript the authors analzyed 152 older adults after a fall from a four wheeled walker.  They described and compared the injury patterns with a special concern for gender-specific patterns. I am not aware of a similar study. The topic is of interest. I have, however, some minor recommendations:

  • Abstract: From the manuscript I get the impression that they included 152 patients from the emergency department, not all patients were admitted. If correct I suggest to make this more clear in the abstract – e.g. add the admission rate, LOS of the hospitalized patients.
  • Methods – secondary outcome: ISS - do you have the information about AIS and ISS? Please add.
  • Results: Did some patients undergo surgery? If yes, please add this information to the results part; what were the reasons for in-hospital death? Please add.
  • Table 2: it remains unclear what a “risk diagnosis” means, please define
  • Table 4: The p-value of the hosp costs seems incorrect, please check
  • There are some typing errors
    • “..” line 26
    • Many hyphenation symbols (“-“) at the wrong position - line 16, 32, 63, 70, 74, 89, 93, 96, 107, 111, 122, 131, 134, 140, 159, 161, 163
    • Some double spaces (“  “) – line 34, 268, 306, 342
    • “,,” line 134
  • Please add the p-value line 33
  • Line 54/55: Does the number of falls belong to falls in Switzerland?
  • Line 93: Please check the spelling “Insel ichospital”
  • Line 25(. Please remove the isolated s between “Rollators s are”

Author Response

We would like to sincerely thank you for your extensive comments, inputs and questions. The review helped a lot to further improve our paper. We hope that the changes meet your requirements.

Since many comments were related to the comparison of our data with international studies, we have written a general statement below:

Abstract: From the manuscript I get the impression that they included 152 patients from the emergency department, not all patients were admitted. If correct I suggest to make this more clear in the abstract – e.g. add the admission rate, LOS of the hospitalized patients.

Answer:

 Thank you for the recommendation. Data regarding admission have now been included in the abstract. As length of stay (LOS) was not statistically significant, we prefer not to mention it in the abstract. However, if there is disagreement, we are willing to address this issue in the abstract of our manuscript.

Methods – secondary outcome: ISS - do you have the information about AIS and ISS? Please add

Answer:

Thank you for your comment regarding AIS and ISS as a secondary outcome measure. Unfortunately, we do not have these data available for the current article. We plan to conduct a larger study in the future that will specifically focus on analyzing these scoring systems (AIS, ISS) in greater detail.

Additionally, our trauma registry database still has some gaps, and not all patients have had these scores calculated. For this reason, we would like to ensure the highest possible data quality and plan to use data validated and analyzed by an independent team from the trauma registry before publication.

Results: Did some patients undergo surgery? If yes, please add this information to the results part; what were the reasons for in-hospital death? Please add.

Answer:

Thank you very much for these suggestions, which are indeed very important with regard to the outcome. We revisited our data and analyzed the patients’ treatment methods, comparing conservative and operative approaches. Following the statistical analysis, we found statistically significant differences (p<0.001) between the female and male groups. The data have been added to Table 4.

Regarding the causes of patient death, we reviewed the final discharge summaries available in the hospital database for patients who died in our hospital. A detailed description of these findings has been included in the “Mortality” section of the Results:

The causes of death in the studied patients reflect a complex interplay between acute injuries and chronic pre-existing conditions, which significantly influenced their health outcomes. Many patients died as a result of traumatic injuries, such as falls leading to skull fractures, subdural hematomas, spinal cord injuries, and other trauma-related complications. In some cases, these injuries were complicated by post-operative delirium, respiratory insufficiency, and sepsis. In addition to acute injuries, the majority of patients had multiple chronic conditions, including cardiovascular diseases (such as coronary artery disease, hypertension, and heart failure), neurological disorders (including dementia, polyneuropathy, and balance disturbances), and metabolic conditions like diabetes and osteoporosis. These underlying health issues often exacerbated the severity of trauma and infections, making recovery difficult and increasing the risk of fatal outcomes. Several patients experienced complications from pre-existing conditions, such as severe heart failure, kidney disease, and chronic respiratory conditions, which directly contributed to mortality. Infections, particularly pneumonia, urinary tract infections, and Clostridium-induced colitis, were also prevalent and played a significant role in the patients' deterioration. Additionally, many patients suffered from impaired mobility, either due to their chronic conditions or as a result of falls, further increasing their vulnerability to life-threatening injuries and complications. Ultimately, the data indicate that the mortality of these patients was heavily influenced by both acute events (such as traumatic falls and surgical complications) and the cumulative burden of chronic health conditions. This highlights the importance of comprehensive medical management, particularly for patients with multiple co-morbidities, to improve their resilience and reduce the risk of fatal outcomes.

Table 2: it remains unclear what a “risk diagnosis” means, please define

Answer:

In our study, the term “risk diagnosis” refers to specific comorbidities or medical conditions that could potentially increase the risk of complications and adversely affect a patient’s rehabilitation or clinical outcome. We grouped such comorbidities into different categories (e.g., musculoskeletal, internal, neurological, psychiatric, ophthalmological, otolaryngological, and geriatric) based on their likelihood of influencing treatment results. These categories were selected because they are frequently cited in clinical practice and literature as contributing to higher morbidity or mortality, particularly in geriatric or polymorbid populations. Therefore, we included “risk diagnoses” in our analysis to capture these additional, potentially significant factors.

This information has been added to the table’s description.

Table 4: The p-value of the hosp costs seems incorrect, please check

Answer:

The p-value for hospitalization costs has been verified in the source data and is consistent in both the test results and the table, with a value of p = 0.125

Line 54/55: Does the number of falls belong to falls in Switzerland?

Answer:

Yes, according Swiss references: Niemann S: Status 2020: Statistik der Nichtberufsunfälle und des Sicherheitsniveaus in der Schweiz. Beratungsstelle für Unfallverhütung BFU 2020.

There are some typing errors

“..” line 26

Many hyphenation symbols (“-“) at the wrong position - line 16, 32, 63, 70, 74, 89, 93, 96, 107, 111, 122, 131, 134, 140, 159, 161, 163

Some double spaces (“  “) – line 34, 268, 306, 342

“,,” line 134

Please add the p-value line 33

Line 93: Please check the spelling “Insel ichospital”

Line 25(. Please remove the isolated s between “Rollators s are”

Answer:

Thank you very much for the thorough spelling corrections. We have done our best to incorporate all of them into the text.

Reviewer 2 Report

Comments and Suggestions for Authors

The authors aim to identify gender differences in older adults who fall while using a rollator. The significance of this aim is not well justified in the introduction of the paper. At a minimum, the authors should describe the known gender differences in injury patterns of fall victims in general, the effectiveness of subsequent interventions, and why the authors suspect this to be significantly different in the rollator falls. This would give the reader a known injury profile for comparison and establish that the results of this study could have clinical impact. Without this, it is difficult to say that this gap in the literature needs addressing.

Methods: Under 2.2 Data handling and extraction, the authors mention, "According to the inclusion and exclusion criteria, the retrieved results were independently checked by two investigators and selected data were validated. The authors should include a measure of agreement between the independent raters, how disagreements were handled, and how the selected data were validated.

Discussion: In several places during the discussion (e.g., lines 303-313, 332-335, 359-360) the authors use non-significant findings to make inferences that are not justified by the study's results. The authors need to address how the statistical differences, medication use, comorbidities, and injury patterns found in the study differ from patterns in general falls and/or the age-matched controls to highlight the significance of the study and draw meaningful conclusions.

Author Response

A special issue of International Journal of Environmental Research and Public Health, special issue : “2nd Edition: Physical Therapy in Geriatrics”

Manuscript ID: ijerph-3383634

Gender-specific patterns of injury in older adults after a fall from a four wheeled walker (rollator): retrospective study from a Swiss level 1 trauma centre

Jolanta Klukowska-Rötzler

Statement Review 2

Dear Reviewer, we appreciate your thorough review of our manuscript and the constructive feedback provided. Below, we address each of your comments and suggestions in detail.

The authors aim to identify gender differences in older adults who fall while using a rollator. The significance of this aim is not well justified in the introduction of the paper. At a minimum, the authors should describe the known gender differences in injury patterns of fall victims in general, the effectiveness of subsequent interventions, and why the authors suspect this to be significantly different in the rollator falls. This would give the reader a known injury profile for comparison and establish that the results of this study could have clinical impact. Without this, it is difficult to say that this gap in the literature needs addressing.

Answer: We thank the reviewer for the valuable recommendation. In response to this insightful comment, we have added a new paragraph in the introduction discussing gender differences in injury patterns among elderly fall victims in general. Additionally, by including the sentence, 'Existing data on gender-related differences in falls among elderly rollator users are limited, but it can be assumed that such differences exist, particularly in relation to biomechanics, device handling, and co-morbidities,' we aim to clarify why we suspect that falls among rollator users may differ from those of elderly fall victims in general.

Methods: Under 2.2 Data handling and extraction, the authors mention, "According to the inclusion and exclusion criteria, the retrieved results were independently checked by two investigators and selected data were validated. The authors should include a measure of agreement between the independent raters, how disagreements were handled, and how the selected data were validated.

Answer:

In accordance with the established inclusion and exclusion criteria, all retrieved results were initially screened by a single investigator. In cases where there was any uncertainty regarding whether to include or exclude a particular record, the investigator consulted with the project leader (senior researcher) to jointly make a final decision. Since only one individual conducted the initial selection, no measure of inter-rater reliability (e.g., Cohen’s kappa) was calculated. The validation process involved discussion and collective analysis of ambiguous cases to ensure consistency with the inclusion and exclusion criteria and to maintain data quality.

Discussion: In several places during the discussion (e.g., lines 303-313, 332-335, 359-360) the authors use non-significant findings to make inferences that are not justified by the study's results. The authors need to address how the statistical differences, medication use, comorbidities, and injury patterns found in the study differ from patterns in general falls and/or the age-matched controls to highlight the significance of the study and draw meaningful conclusions.

Answer:

We thank the reviewer for the helpful comment. In response to her/his recommendation, we have removed some data that were statistically not significant. Additionally, we have included data on falls in older adults in general. While we acknowledge that co-morbidities and prior medication were not statistically significant in our study, we consider them important risk factors for falls in the elderly and have therefore retained their discussion in the manuscript. However, if there is disagreement, we are willing to remove these sections from the discussion. Regarding injury patterns, we believe they are thoroughly addressed, including comparisons with falls in geriatric patients in general, and thus no further changes were made. Nevertheless, if needed, we are open to revising this part of the discussion. Finally, we have included the significant finding that operative procedures were statistically more frequent in women, and this has been clearly stated in the manuscript.

Reviewer 3 Report

Comments and Suggestions for Authors

The authors are to be commended for their work. This study addresses a significant challenge facing societies as they age, offering insights with broad public health and planning implications. 

The objective of the study was to elucidate gender-specific patterns of injury associated with rollators (or four-wheeled walkers).

The introduction is comprehensive and provides a summary of the existing literature on the subject matter. The Materials and Methods section provides a detailed account of the features of the study. The inclusion and exclusion criteria are clearly delineated. 

In the Results section, the findings are elucidated and corroborated with the aid of tables and visualisations.

All significant findings are highlighted in the discussion section. At the conclusion of the study, the initial research question is clearly addressed.

The references are current and pertinent to the subject matter under investigation.

Following the correction of a few typographical errors, the study is suitable for publication.

There are numerous spelling errors, such as those observed in lines 16, 26, 63, 76, and 93. A language review would be beneficial.

Comments on the Quality of English Language

There are numerous spelling errors, such as those observed in lines 16, 26, 63, 76, and 93. A language review would be beneficial.

Author Response

A special issue of International Journal of Environmental Research and Public Health, special issue : “2nd Edition: Physical Therapy in Geriatrics”

Manuscript ID: ijerph-3383634

Gender-specific patterns of injury in older adults after a fall from a four wheeled walker (rollator): retrospective study from a Swiss level 1 trauma centre

Jolanta Klukowska-Rötzler

Statement Review 3

Comments and Suggestions for Authors

The authors are to be commended for their work. This study addresses a significant challenge facing societies as they age, offering insights with broad public health and planning implications. 

The objective of the study was to elucidate gender-specific patterns of injury associated with rollators (or four-wheeled walkers).

The introduction is comprehensive and provides a summary of the existing literature on the subject matter. The Materials and Methods section provides a detailed account of the features of the study. The inclusion and exclusion criteria are clearly delineated. 

In the Results section, the findings are elucidated and corroborated with the aid of tables and visualisations.

All significant findings are highlighted in the discussion section. At the conclusion of the study, the initial research question is clearly addressed.

The references are current and pertinent to the subject matter under investigation.

Following the correction of a few typographical errors, the study is suitable for publication.

There are numerous spelling errors, such as those observed in lines 16, 26, 63, 76, and 93. A language review would be beneficial.

Answer 3

Dear Reviewer,

Thank you for your thoughtful and encouraging feedback on our manuscript. We are grateful for your recognition of the significance of our study and its implications for public health and planning in aging societies. Below, we address your specific comments and suggestions.

We appreciate your observation regarding typographical errors. A thorough language and typographical review has been conducted across the entire manuscript to correct all identified errors and ensure clarity and consistency. Specific corrections have been made to the sections you referenced, including the noted lines. Additionally, we engaged a professional language editing service to further refine the manuscript.
